# Emerging Technologies for Prolonging Fresh-Cut Fruits' Quality and Safety during Storage

Rey David Iturralde-García, Francisco Javier Cinco-Moroyoqui [ID], Oliviert Martínez-Cruz, Saúl Ruiz-Cruz, Francisco Javier Wong-Corral, Jesús Borboa-Flores, Yaael Isbeth Cornejo-Ramírez [ID], Ariadna Thalia Bernal-Mercado *[ID] and Carmen Lizette Del-Toro-Sánchez *[ID]

Departamento de Investigación y Posgrado en Alimentos, Universidad de Sonora, Blvd. Luis Encinas y Rosales S/N, Col. Centro, Hermosillo 83000, Mexico
* Correspondence: thalia.bernal@unison.mx (A.T.B.-M.); carmen.deltoro@unison.mx (C.L.D.-T.-S.)

**Abstract:** Fresh-cut fruits have been in great demand by consumers owing to the convenience of buying them in shopping centers as ready-to-eat products, and various advantages, such as the fact that they are healthy and fresh products. However, their shelf lives are brief due to their physiological changes and maturation. Therefore, this review includes information from the physicochemical, microbiological, nutritional, and sensory points of view on the deterioration mechanisms of fresh-cut fruits. In addition, updated information is presented on the different emerging technologies, such as active packaging (edible films, coatings, and modified atmospheres), natural preservatives (antioxidants and antimicrobials), and physical treatments (high hydrostatic pressure, UV-C radiation, and ozone). The benefits and disadvantages of each of these technologies and the ease of their applications are discussed. Having alternatives to preserve fresh-cut fruit is essential both for the consumer and the merchant, since the consumer could then obtain a high-quality product maintaining all its properties without causing any damage, and the merchant would receive economic benefits by having more time to sell the product.

**Keywords:** microbial quality; sensory quality; edible coating and films; natural antioxidants; natural antimicrobials; modified atmospheres; UV-C; ozone; high hydrostatic pressure

## 1. Introduction

In recent years, fruit consumption has increased in a large portion of the population due to the current concern for a healthy lifestyle [1]. Fruits are a rich source of vitamins, minerals, and dietary fiber, which are essential for the human diet. Due to a population that is very busy with their different activities, and which consequently has less time to prepare its food, there has been a tendency to demand fresh, nutritious, convenient, and quickly accessible products, such as fresh-cut produce. Therefore, fresh-cut fruits constitute one of the fastest-growing food industry segments [2]. The Food and Drug Administration (FDA) defines fresh-cut fruit as any fruit that has been changed physically from its natural state by minimal processing, such as chopping, dicing, peeling, shredding, slicing, spiralizing, or tearing [3]. Fresh-cut fruits remain fresh without additional treatment, such as blanching, freezing, canning, cooking, or adding juice, syrup, or dressing. These products can be considered ready-to-eat but may or may not be washed before being packed for use by the customer or a retail food store. Fresh-cut fruits can include single or mixed fruits in the same packaging, providing great convenience, nutritional value, taste, and freshness [3].

The maintenance of the quality of fresh-cut fruits is attributed to the physiological and biological mechanisms of each fruit and environmental conditions such as storage conditions, temperature, and humidity [4]. The main disadvantage of fresh-cut fruits is their short shelf-lives, often less than two weeks [5]. When fresh fruits are cut or minimally processed, they are susceptible to chemical, physical, microbiological, and sensory changes.

This affects the product's marketability and reduces its nutritional value, and several food-borne outbreaks are linked to fresh-cut produce [1]. Additionally, it must be considered that many other factors can impact the final product, such as the type of packaging, postharvest processing, and type of cultivar, among others [6]. Whole fruits have a natural barrier (peel) that protects them from spoilage, microorganisms, or environmental conditions; however, this barrier is typically removed during the processing of fresh-cut fruits, making them more susceptible to decay (Figure 1) [7]. It is even more challenging to maintain the quality of fresh-cut fruits than vegetables due to their complicated physiology [8].

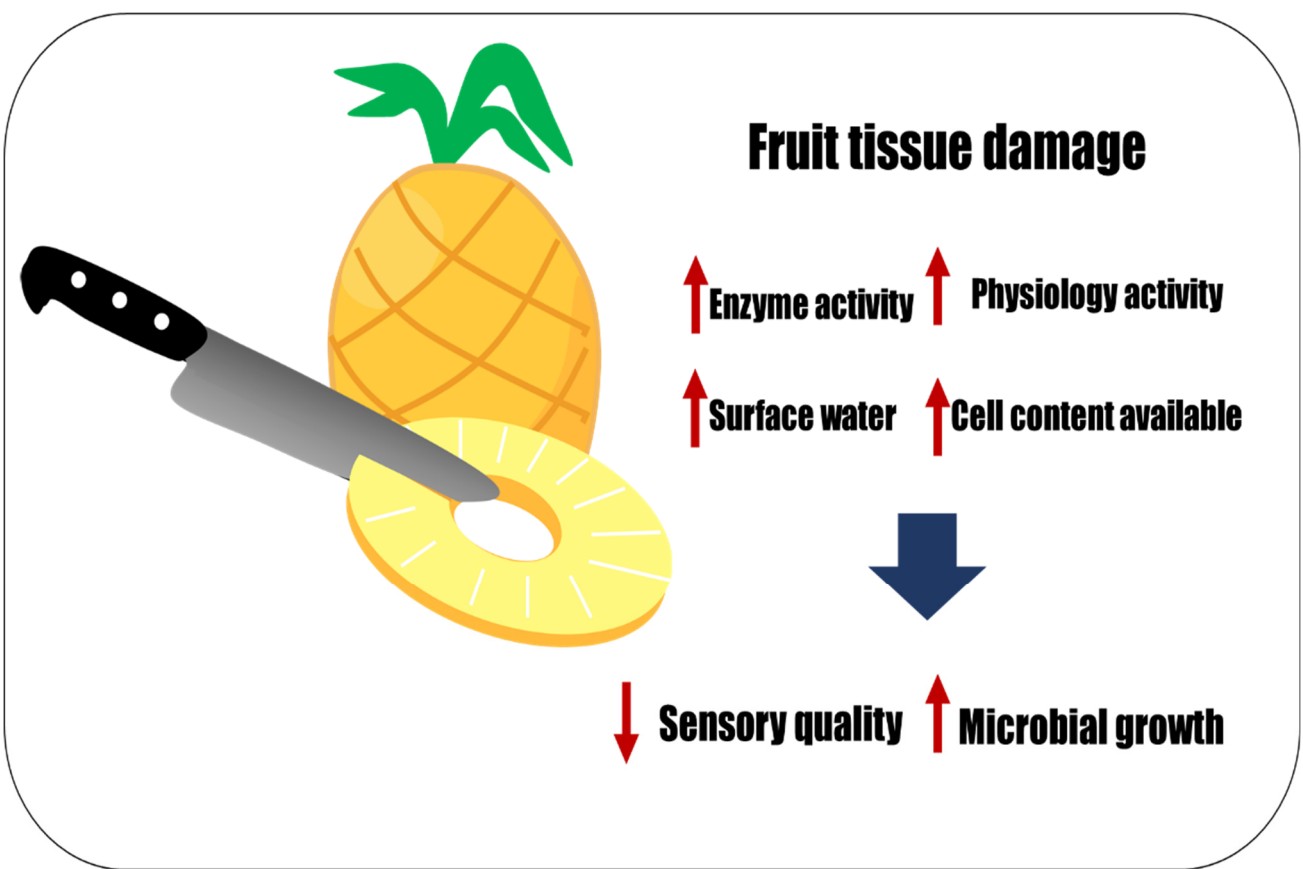

**Figure 1.** Minimal processing affects sensory quality and promotes microbial growth. The red up arrow indicates increase and the red down arrow indicates decrease.

Several methods could be employed to preserve fruits; however, as fresh-cut fruits are consumed fresh, thermal or freeze techniques are unsuitable because they impair the sensory, nutritional, and physicochemical quality. As a result, finding effective strategies to prevent microorganism growth while maintaining the quality of fresh-cut fruits is of great interest to food industries. The conventional way of retaining the fresh-cut products' microbiological quality is through washing and disinfection with chlorine due to its low cost and ease of use. However, it has been hypothesized that chlorination's disinfection byproducts may pose a health risk to humans [9]. Moreover, consumers demand more natural and fresh products without synthetic additives, making them nutritionally healthier. Therefore, technologies such as active packing, natural additives, ultraviolet light, high hydrostatic pressure, and ozone treatments are emerging to improve fresh-cut fruits' microbiology and sensory quality. This review summarizes some recent findings concerning the use of emerging technologies to improve the quality of fresh-cut fruit.

Our article summarizes recent studies about the main alternatives with which to maintain the sensory physicochemical quality and ensure the microbiological safety of fresh-cut fruits. In the last year, several reviews have addressed the issue of fresh-cut fruits'

preservation with some differences. For example, Yousuf et al. [10] conducted a critical study to summarize the impacts of utilizing essential-oil edible coatings on fresh/fresh-cut fruits and vegetables. The primary emphasis was on the inclusion of essential oils in edible coatings, their benefits and drawbacks, methods of extracting essential oils, and the results obtained from their use in fruits, fresh cut fruits, and vegetables to extend their shelf lives. The key difference between our study and theirs is that they only focused on a single technology, whereas we summarize multiple conservation strategies. Giannakourou and Tsironi [1] outlined the hurdles of technology applied in the preservation and shelf-life extension of fresh-cut fruits and vegetables. This review highlights the combinations of traditional methods, such as temperature control and chemical agents, and a number of emerging technologies, including high hydrostatic pressure, UV-C irradiation, pulse light, ozone, ultrasound, and some packaging methods. The primary goal of this study was to emphasize the combination of technologies to boost efficacy; it also reviews studies on entire fruits and vegetables in comparison to our study.

Other reviews individually addressed different technologies; for example, Botondi, Barone, and Grasso [2] in their review aimed to increase understanding of environmentally beneficial technologies, such as ozone, which prolongs shelf life and preserves the quality of fresh-cut fruits and vegetables without emitting toxic chemicals that can harm plant material and the environment. Zhang et al. [11] highlighted the most recently published studies based on the use of plant extracts for browning suppression in fresh-cut fruits and vegetables. They described the types of plant extracts, their anti-browning capabilities, the main extraction techniques, and application procedures for fresh-cut fruits and vegetables. Kocira et al. [12] shows the application, trends and perspectives of polysaccharide coatings and edible films, and their influences on the quality of fruits and vegetables, while demonstrating their main functions and advantages. However, this review does not consider fresh-cut fruits.

## 2. Fresh-Cut Fruit Processing Impacts Physicochemical, Sensory, and Microbial Quality

When fresh fruits are cut or minimally processed (peeling, cutting, or slicing), they are susceptible to chemical, physical, sensory, and microbiological changes. Wounding during processing results in an increase in off-flavor compounds, loss of firmness and respiration, reduced fresh-cut shelf life, and senescence processes [4]. One of the degradation factors of these products is the color change. For the consumer, the product's first appearance is significant, since each product's characteristic color indicates its freshness and quality [13]. Polyphenol oxidase (PPO) and peroxidase (POD) are the main enzymes that cause color degradation and degradation of other sensory properties, such as taste and flavor, in fresh-cut fruits [14]. In particular, the PPO enzyme catalyzes the oxidation of polyphenols to o-quinones in the presence of oxygen, producing undesirable pigments (brown coloration of many fruits and vegetables) during ripening, storage, processing, and handling. At the same time, the POD enzyme (considered antioxidant) catalyzes the conversion of hydrogen peroxide to water using various substrates, such as polyphenols, lipids, or other compounds [15]. There are several studies focused on the inhibition of these enzymes using different inhibitory compounds, such as volatile compounds, cysteine, and ascorbic acid [16,17]; or by other processes, such as microwaving [18], ultrasonication [19], high isostatic pressure, and thermal conditions [20,21].

Flavor and texture are other attributes that must be considered in consumer satisfaction. The flavor is commonly related to aroma (odor) and taste (salty, sour, bitter, sweet, and umami). Umami is described by salts of amino acids and nucleotides [22]. The flavor in fresh-cut fruits can be affected by the increases in sugar contents, decrease in organic acids, and changes in aroma (volatile compounds) [13]. Enzymes such as lipoxygenase or peroxidase lead to the development of undesirable flavors—rancid, cardboard, or oxidized off-flavors (Figure 2) [23].

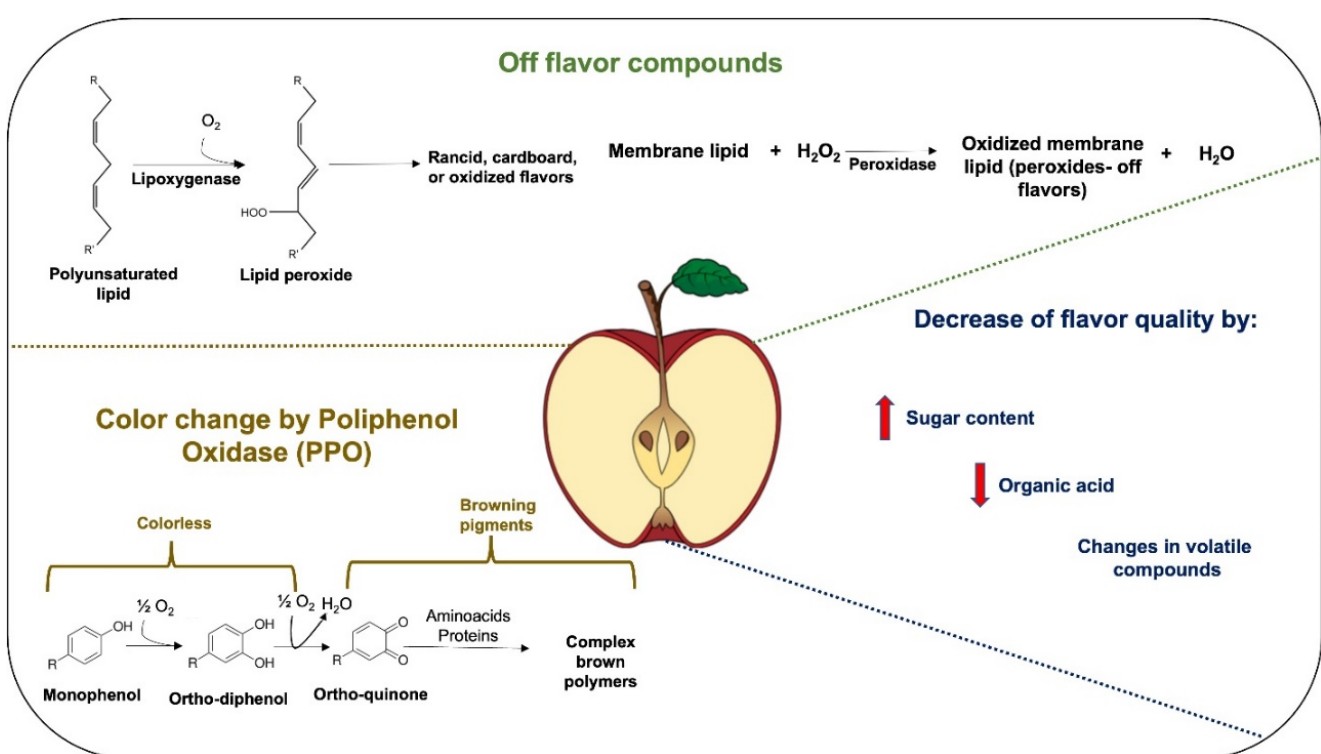

**Figure 2.** Enzymatic reactions and other factors that produce browning and off-flavors of fresh-cut fruits. The red up arrow indicates increase and the red down arrow indicates decrease.

On the other hand, the textural properties of fresh-cut fruits are related to the deformation, disintegration, and flow of the food under a force. The texture is related to the cell wall components (pectin, hemicellulose, and cellulose), generating the softening of fresh-cut fruits during storage by enzymatic or non-enzymatic mechanisms. This is due to the loss of cohesiveness, which decreases intermolecular bonding between cell wall polymers. Consequently, an increase in the solubility of the wall constituents is observed, principally pectin, generating the softening. Other factors, such as pH, salt concentration, and water, deficiency could influence texture loss [24].

Salinas-Hernández et al. [25] predicted fresh-cut mango's shelf life, indicating that physicochemical variables related to sensory changes can be identified and used as quality indicators. Therefore, physicochemical variables (weight, volume, pH, titratable acidity, soluble solids content, volatiles) with high correlations with the sensory attributes (color, texture, and flavor) can be used as indicators of the sensory changes, since they are easy to measure, objective, and relatively inexpensive. Additionally, the identification of these variables was carried out by regression analysis. There are several studies relating physicochemical and sensory characteristics of papaya [26], persimmon [27], plums [28], mango-based fruit bar [29], jackfruit [30], apples [31], and oranges [32], among other fresh-cut fruits.

Fresh-cut fruits are considered perishable due to their intrinsic characteristics and minimal processing that favor microbial growth, which can cause changes in safety and sensory and physicochemical properties. Fresh-cut fruit cells naturally deteriorate with time and are influenced by conditions after harvest, processing, and storage. Peeling and processing cause physiological damage and cell disruption, resulting in the release of their components, such as carbohydrates and proteins, which serve as a source of nutrients for native or exogenous microorganisms [33]. Minimal processing destroys living plants' protective membranes and barriers, allowing microbial pathogens to enter and contaminate them. The chances of food-borne illness due to pathogens or spoilage organisms growing in these products are very high. In addition, fresh-cut fruits have water activity levels between 0.8 and 0.99 and pH values between 3.0 and 7.5, suitable conditions for the

growth of microorganisms [2]. The types and growth rates of the microorganisms will be significantly influenced by the product's temperature, the relative humidity, the atmosphere, and intrinsic factors such as pH, water content, and nutrients [34].

Fresh-cut fruits are prone to the invasion and colonization of microorganisms such as mesophiles and psychrophiles, molds, and yeasts [2]. Two types of microorganisms can contaminate these products: deteriorative and pathogenic. Deteriorative microorganisms cause damage to fresh-cut fruits and make them sensorily unacceptable. For example, this type of microorganism causes the production of lactic acid, acetic acid, hydrogen gases, and carbon dioxide, which results in sour odors and puffing up of packages. Other products include thiols, esters, amines, and peroxides that cause off-flavors, odors, and color changes. In addition, spoilage microorganisms can produce enzymes such as proteases, lipases, and amylases that cause structural changes in tissue and flavor [34]. The incidence of pathogenic microorganisms represents a food safety risk. Fresh-cut fruits when consumed raw are vehicles for the transmission of pathogens. Among the most common pathogenic microorganisms found in fresh-cut fruits are bacteria such as *Listeria monocytogenes*, *Escherichia coli*, *Salmonella enterica*, *Campylobacter* spp., and *Staphylococcus aureus*; fungi such as *Alternaria* spp., *Penicillium* spp., *Botrytis* spp., *Rhizopus* spp., and *Colletotrichum* spp.; and some viruses, such as norovirus and hepatitis A [2,35–37]. These microorganisms are responsible for many outbreaks worldwide [37]. The microbiological safety of fresh-cut fruits is crucial to maintaining their commercial value. Microbial contamination has an economic impact on food loss because it reduces the product's shelf life through deterioration and risks the public's health by causing foodborne illnesses [33,38]. As these products are consumed without any thermal treatment, it is essential to keep the microbial loads of fresh-cut fruits as low as possible to avoid foodborne diseases.

One of the key elements influencing the quality of fresh-cut fruits is exposure to cold temperatures immediately after harvest to reduce the effects of cutting stress. Sometimes a temperature just above that which would cause chilling injury gives the optimal condition for quality. Due to the perishable nature of fresh-cut fruits, on some occasions, it is preferable to store them under refrigeration at a temperature that could cause slight chilling injury as opposed to one that would promote quick natural deterioration [39]. Chilling damage to fresh-cut fruits can have various symptoms, some of which are noted despite minor visual manifestations; for example, poor flavor retention associated with the inhibition of volatile aroma production, increased respiration rates in fresh-cuts relative to the corresponding whole fruit, and tissue transparency and juice leakage because membrane damage, particularly in fresh-cut tropical and subtropical fruit. The signs of chilling injury in the entire fruit can also be generalized for fresh-cut products, such as softening or other textural alterations, pigment loss, and increased $CO_2$ production [40]. In fresh-cut fruits, chilling injury symptoms are caused mainly by the physical shifting of the membrane from a liquid–crystalline to a solid–gel phase during chilling. This process is highly reliant on the degree of membrane lipid saturation. The fluidity-dependent membranes start solidifying at cool temperatures, causing membrane integrity/leakiness problems, solute diffusion, tissue water loss, and membrane-bound protein agglomeration. Due to disruptions in the membranes connected to the electron transport chain, suboptimal chilling temperatures may also hasten oxidatively induced senescence and increase the accumulation of active oxygen species [39].

Cold storage temperatures have an impact on final product quality. For example, fresh-cut mango, a tropical fruit, is susceptible to chilling injury when stored at low temperatures, compromising its overall sensory quality. The ideal storage temperature for fresh-cut fruits is never more than 5 °C, a chilling temperature for freezing delicate tropical fruits such as mangoes. Despite the possibility of chilling injury, Dea, Brecht, Nunes, and Baldwin [40] showed that fresh-cut mango slices had a longer shelf-life when stored at 5 °C than at 12 °C because the negative changes in appearance and aroma that occurred at the higher temperature were more unpleasant than the moderate negative changes that appeared at a lower temperature. Marrero and Kader [41] reported that temperature was the primary



factor affecting the quality of fresh-cut pineapple. The pulp pieces' post-cutting life was 4 days at 10 °C, yet more than 2 weeks at 0 °C. This increased longevity at levels below the chilling injury limit is also the case for whole fruits. At all temperatures, a rapid increase in respiration followed by an increase in ethylene production marked the end of commercial life. Beyond this point, storage was continued, which resulted in the development of off-tastes, smells, and microbiological deterioration. Even after being maintained at 0 °C for two weeks, the pulp fragments exhibited no symptoms of chilling damage. Since some of the symptoms may not appear for several days after being transferred to non-chilling temperatures, this may be because of the short time (approximately 3 h) between removal from refrigerated storage and quality evaluation. The impacts of temperature on fresh-cut fruits' physicochemical, sensory, and microbiological quality still require more research.

## 3. Emerging Technologies to Preserve the Shelf Lives of Fresh-Cut Fruits

### 3.1. Active Packaging

#### 3.1.1. Edible Films and Coatings

Some technologies have been developed to preserve fresh-cut fruits, such as edible packaging [42–44]. This includes any edible material used to wrap food to prolong its shelf life that can be safely consumed with it. Edible films and coatings present exciting features, including biocompatibility, edibility, and a wide range of applications, making them excellent alternatives for fruit preservation [45]. The terms "film" and "coating" are frequently used interchangeably to describe a relatively thin layer of edible material covering a product's surface. While a coating is placed and formed directly on the food's surface, a film is sometimes distinguished from a coating because it is a stand-alone wrapping material. Edible coatings are typically used for liquid applications, whereas edible films are used as solid sheets and subsequently applied to food products [46].

Edible films and coating materials must be safe for human consumption, since they are not removed before consuming the product and must not alter the original product's taste, texture, smell, or appearance [47]. These emerging technologies can address customers' demands for more natural, nutritious, ready-to-eat, and minimally processed products without generating waste [48]. In addition, edible films and coatings may replace, to some extent, plastic packaging with natural and biodegradable substances. Therefore, their use could significantly reduce packaging requirements and waste disposal problems [49].

Edible films are generally good moisture barriers, thereby restricting moisture exchange between fresh-cut fruits and the atmosphere, hence reducing microbial development, weight loss, texture changes, and undesired chemical and enzymatic reactions. Fresh-cut fruits experience less respiration and senescence due to the changing environment caused by edible components' good oxygen and gas barriers [50]. Furthermore, coatings may improve the visual quality by providing gloss to the coated commodities [51]. Edible films and coatings can also be used as carriers of antioxidants, flavoring agents, coloring agents, growth regulators, and antimicrobials that will improve food quality and safety [52–54].

Coating materials include carbohydrates, proteins, lipids, and combinations [55]. Polysaccharides such as alginate, pectin, cellulose, starch, chitosan, carrageenan, arabic gum, and xanthan gum are polysaccharides that constitute the fundamental structure of the polymeric matrix [56–58]. These polymers are also effective gas barriers at low and intermediate relative humidity levels because they are hydrophilic, but due to their high water vapor permeability, they are poor moisture barriers [59]. Paraffin, carnauba, beeswax, shellac resin, and certain oils are the most popular lipids used for edible coatings because they have strong barrier qualities [60]. Proteins such as milk casein, milk zein, maize, and whey are beneficial as gas barriers ($O_2$ and $CO_2$) and antimicrobial carriers in coating treatments; however, they have limited water barrier capabilities [61,62]. Plasticizers, stabilizers, and emulsifiers can also be used to improve edible coatings' physical and chemical characteristics [60]. The functionality of covers and films is commonly evaluated through mechanical properties, such as strength, elasticity, and rheological properties [46].

Edible coatings are applied using various methods, such as immersion (dipping), spraying, and brushing (Figure 3) [46]. The dipping process submerges the product in the coating solution for a period and then lets it dry for a few minutes. For fresh-cut fruits, the immersion approach is the most popular [63]. The spraying direction is appropriate when the coating solution has a low viscosity and can be sprayed over the product. Brushing involves applying the coating solution directly to the product's surface with a brush; nevertheless, some variables, such as the amount left in the brush, are difficult to control and may influence this process [64].

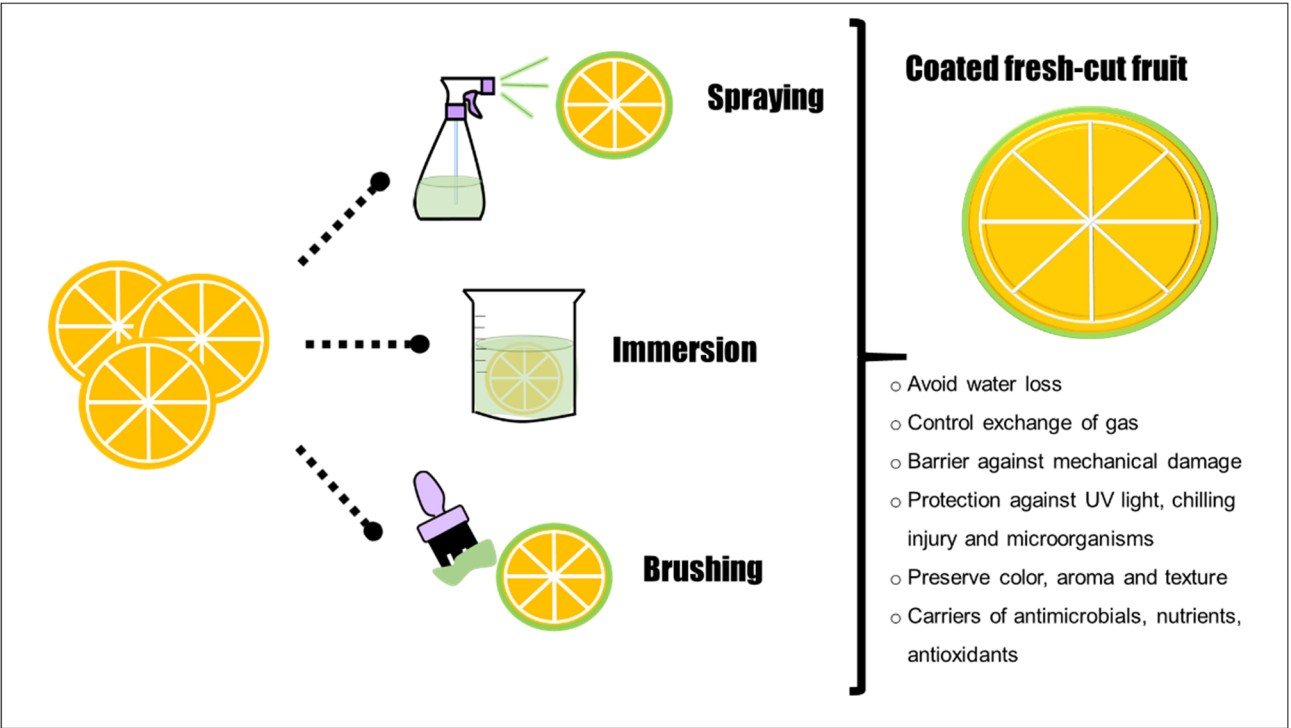

**Figure 3.** Main goals of edible coatings and how they are applied to fresh-cut fruits.

The loss of quality of fresh-cut fruits is due to physiological changes in the fruits, such as an altered respiration rate, loss of water, changes in texture, a decrease in organic acids, an increase in soluble solids, and starch breakdown, among others. Numerous studies have been conducted to study edible coatings in various fruits to avoid physiological changes and prolong their shelf lives (Table 1), considering the appropriate coating materials [4]. During postharvest deterioration, fresh fruits are particularly prone to weight loss, which contributes to product wilting and loss of textural qualities, such as softening and crispness, resulting in low market value and poor consumer acceptance [4]. In this context, adding ascorbic acid (1%) to a chitosan-based coating on strawberry fruits during cold storage for 15 days reduced weight loss. It also suppressed fruit softening by reducing cell wall degrading enzymes, such as polygalacturonase, cellulase, and pectin methyl esterase activities [65]. Furthermore, this treatment reduced the percentage of decayed fruits while maintaining soluble solids, titratable acidity, and antioxidant activity without affecting the fruits' sensory quality (color, taste, glossiness, and overall acceptability).

Respiration rate is one element that contributes to postharvest loss and the shelf lives of fresh-cut fruits [10]. Edible coatings can reduce respiration by establishing a modified internal atmosphere that acts as a barrier to oxygen and carbon dioxide. Afifah et al. [66] studied how the respiration rate of fresh-cut apple slices coated with chitosan and stevia combinations changed after three days of storage in modified atmosphere packing (polypropylene-PP, 30 μm) at +1 °C. The edible coatings reduced the respiration rate, which was explained by the diffusion of oxygen from the packaging atmosphere to the

fruit tissue, due to the high $O_2$ resistance generated by the edible coating, which slowed plant metabolism.

**Table 1.** Edible coatings and films used to improve fresh-cut fruits' quality.

| Fresh-Cut Fruit | Edible Film/Coating Material | Treatment Application | Storage Condition | Results | Reference |
|---|---|---|---|---|---|
| Apple | Sodium alginate + Tween-80 + glycerol + thymol-ethanol solution | The container was covered | 5 days at 4 °C | • Increase antioxidant status<br>• Inhibition of *S. aureus* and *E. coli* growth<br>• Reduction of weight loss<br>• Retain nutrition and surface color | [67] |
| Apple | Whey protein concentrate + apple pomace extract | Immersion | 12 days at 5 °C | • Decrease in the weight loss<br>• Reduction of browning index<br>• Reduction of microbial loads<br>• Slight effect in sensory evaluation | [68] |
| Apple | Sodium alginate + carboxymethyl cellulose + glycerol + calcium chloride + citric acid + shallot waste extracts | Wrapped films | 12 days at 4 °C | • Prevention of browning index<br>• Maintain the overall quality | [69] |
| Apple | Chitosan + ascorbic acid | Immersion | 14 days at 5 °C | • Suppress browning<br>• Retain flesh firmness<br>• Maintain phenolic compounds throughout<br>• Delay the microbial growth | [70] |
| Apple | Pectin + whey protein + sweet orange essential oil or lemon essential oil | Immersion | 7 days at 4 °C | • Reduce weight loss<br>• Reduce changes in color and texture<br>• Reduce microbial counts<br>• No change in flavor | [71] |
| Apple | Chitosan + gelatin + tannic acid | Cover for polyethylene terephthalate packages box | 10 days at 4 °C | • Decrease weight loss<br>• Delay browning degree<br>• Inhibit the lipid oxidase activity<br>• Decrease the malondialdehyde content | [72] |
| Kiwifruit | Aloe vera gel + hydroxypropyl methylcellulose + lemon essential oil | Spraying | 10 days at 4 °C | • Act as a barrier to gas exchange<br>• Reduce microbial load<br>• Present Herbaceous and lemon taste<br>• Preserve the quality of fresh-cut kiwifruit | [73] |

<div align="center">

**Table 1.** *Cont.*

</div>

| Fresh-Cut Fruit | Edible Film/Coating Material | Treatment Application | Storage Condition | Results | Reference |
|---|---|---|---|---|---|
| Mango | Carrageenan + beeswax | Immersion | 6 days at 6 °C | • Reduce weight loss <br> • Delay an increase in total soluble solids <br> • Maintain pH, total acidity, and growth of microorganisms. <br> • No difference in sensory perception | [66] |
| Mango | Citric, ascorbic + potassium sorbate acid + aloe vera | Immersion | 6 days at 7 °C | • Resist the loss of water <br> • More weight shrinkage <br> • Retain the brightness <br> • Inhibit the development of redness <br> • Maintain the yellow color <br> • Maintain vitamin C levels | [74] |
| Melon | Citral nanoemulsions + chitosan or carboxymethyl cellulose | Immersion | 7 °C for 14 days | • Reduce microbial counts antimicrobial protection <br> • Extent product's storability | [75] |
| Orange | Sodium alginate + cocoa | Immersion | 9 days at 6 °C | • Maintain texture quality <br> • Maintain the microbiological properties (yeast and mesophilic aerobic bacteria) <br> • Maintain the sensory properties | [32] |
| Papaya | Alginate + oregano essential oil | Immersion | 12 days at 4 °C | • Reduce water loss <br> • Increase sensory scores with low oil concentrations | [76] |
| Papaya | Starch + stearic acid + aloe vera | Immersion | 12 days at 10 °C | • Reduce weight loss <br> • Improve firmness <br> • Retain color <br> • Slight decrease in ascorbic acid <br> • Extent microbial quality | [77] |
| Pear | Whey protein | Immersion | 4 °C for 28 days | • Reduce browning <br> • Maintain firmness <br> • Maintain taste, smell, color, and hardness properties | [43] |
| Pineapple | Sodium alginate + citral nanoemulsion | Immersion | 4 °C for 12 days | • Improve color retention (higher L* and b* values) <br> • Reduce respiration rate <br> • Reduce microbial growth (*S. enterica* and *L. monocytogenes*) <br> • Higher concentrations of citral cause a decrease in texture and sensory acceptance. | [78] |

**Table 1.** *Cont.*

| Fresh-Cut Fruit | Edible Film/Coating Material | Treatment Application | Storage Condition | Results | Reference |
|---|---|---|---|---|---|
| Strawberry | Alginate + calcium chloride | Immersion | 15 days at 4 °C | • Reduce respiration and transpiration rates<br>• Delay the increase in the pH and soluble solid content<br>• Delay surface mold growth<br>• Preserve the sensory properties (color and texture) | [79] |

L* = brightness; b* = yellow/blue coordinates.

Citric acid, malic acid, and glutamic acid are the primary sources of titrable acidity in fresh-cut fruits and serve as substrates for respiration; therefore, titrable acidity drops with fruit ripening or maturity. By reducing respiration, edible coatings effectively postpone the loss of organic acids. For example, Alharaty and Ramaswamy [79] coated fresh-cut strawberries, and they showed higher citric acid (%) than the uncoated samples. The acidity decreased slightly in the coated samples during the 15 days of storage compared to uncoated fruits. On the other hand, the breakdown of starch into soluble sugars or the hydrolysis of cell walls increases the total soluble solids of fruits during storage. By limiting the respiration rate, the edible coating reduces the breakdown of complex sugars into simple sugars. In this approach, fresh-cut Fuji apples were treated with three edible coatings based on Aloe vera gel and lemon essential oils applied by the spraying method [80]. The results showed that soluble solids content was reduced in uncoated samples, and the treatment was able to limit the loss of soluble solids content from the first three days of storage by maintaining this behavior until the last day of evaluation. In addition, this edible coating reduced weight loss and color changes and delayed the browning process during cold storage without affecting apples' taste, aroma, or flavor. The total soluble solids allow for the measurement of sugar content, and when combined with titratable acidity, provide helpful input on consumer satisfaction with the fruit.

The potential for developing edible antimicrobial coatings as a cost-effective way to extend the shelf lives of fresh-cut fruit has been examined. Antimicrobial chemicals can be delivered through edible films and coatings to prevent food spoiling caused by bacteria, extending product shelf life by many folds [75]. Antimicrobial compounds can be found naturally in the raw material used to make the packaging, or they can be added to it specifically. Some antimicrobial chemicals have been isolated from plants and tested in various polymeric matrixes for antibacterial activity [81]. Zhang et al. [82] produced sodium alginate/pullulan composite films with different capsaicin concentrations. Their physicochemical characteristics were studied thoroughly, and the film's transmittance, elongation at break, and moisture content dropped as the capsaicin concentration increased. In contrast, the tensile strength, water vapor permeability, and surface contact angle increased. Furthermore, the composite films showed good antibacterial activity against *E. coli* and *S. aureus* in liquid culture tests. Different composite films covered fresh mature apple cubes. As the capsaicin content increased in the film, the apples stayed fresh longer, which demonstrated that the edible films could prolong the shelf life of apples and inhibit the growth of bacteria.

Organic acids (acetic, benzoic, lactic, propionic, sorbic), fatty acid esters (glyceryl monolaurate), polypeptides (lysozyme, peroxidase, lactoferrin, nisin), plant essential oils (EOs) (cinnamon, oregano, lemongrass), nitrites, and sulfites are among the antimicrobials that can be incorporated into edible coatings [46]. In a recent study, Marquez et al. [83] coated fresh-cut apples with a blended whey protein/pectin film made in transglutaminase, and the coated and untreated fruits samples were compared throughout 10 days of stor-

age. The treatment-preserved phenolic content prevented microbial growth, and reduced hardness and chewiness without affecting sensory acceptability. All these studies prove the potential of edible coatings to keep fresh-cut fruits' physicochemical, microbiological, and sensory quality. However, there is still a lack of studies that handle more samples and are applied in real conditions.

Even though various edible coatings and films have been successfully deposited on fresh-cut fruits, this technique may have some limitations and negatively impact the final product's quality. For example, a thick coating on the surface of fruit creates an unfavorable barrier between the interior and exterior atmosphere and prevents the exchange of respiratory gases ($CO_2$ and $O_2$), lowering the fruit's quality [84]. Therefore, it is essential to modify the coating's thickness for the variety, storage conditions, and marketing temperatures. Additionally, for edible coatings, the type of material should be considered based on whether the fruit is climacteric or not to cover the fruit's needs, delay ripening, and reduce weight loss. Incorporating antimicrobials, antioxidants, nutraceuticals, or other substances into the coatings could give fresh-cut fruits an undesirable odor, flavor, or color, mainly if essential oils are employed [85]. Additionally, the incorporation of bioactive compounds can affect the mechanical properties of the edible coatings [86]. This must be contemplated before its application to assure sensory quality and consumer acceptance. Another restriction may be increased cost, since more materials and operations were added to the fresh-cut fruits [85]. Therefore, it is crucial to search for low-cost materials and employ highly effective manufacturing and application techniques. Another drawback found in edible coatings is that they must be regulated and declared on the labels; and it must be ensured that the material used and added ingredients are non-toxic, food grade, and meet the highest hygiene standards [86].

### 3.1.2. Modified Atmosphere Packaging

Modified atmosphere packaging (MAP) is an inexpensive preservation technique used to extend the shelf life of fruit postharvest by slowing its respiration rate and senescence, and inhibiting microbial growth [87,88]. This technology reduces food waste and constitutes 12.3 of the United Nations' 2030 Sustainable Development Goals [89,90].

Active MAP is a method that involves the removal of air from inside the package (by flushing or evacuation-backflushing) and its replacement by a gas or gas mixture ($N_2$, $CO_2$, and $O_2$) supplied from pressurized cylinders or otherwise [87,91]. Therefore, applying MAP requires packing which is airtight to maintain gas concentrations during the necessary exposure time [92]. Another advantage of this package type is the biogeneration of a MA: the commodity of producing the MA by reducing the $O_2$ and increasing the $CO_2$ levels through respiratory metabolism [88]. Packaging permeability is another factor that influences gaseous composition. Gaseous composition can be modified by micro-perforations in the film (commonly 60–120 µm), and the size, shape, and method of hole production are crucial for MAP's effectiveness [88].

Oxygen and carbon dioxide are the most important gasses, because these two gases are crucial parts of the respiration processes in fruits. These gases inside the package decrease the respiration rate affecting all the fresh-cut fruits' metabolic pathways. Elevating the carbon dioxide concentration at the expense of oxygen decreases the respiration rate [91].

The reduction of the respiration rate with MAP support prevents oxidation and reduces ethylene biosynthesis. Oxidation is a factor that induces the browning in fruits and leads to the loss of the nutritional value of fruits due to the destruction of many nutrients, such as vitamins and proteins [93]. Fresh-cut fruits contain phenolic compounds that prevent staining or discoloration when processed into fresh-cut products. This, combined with MAP, allows enzyme-mediated browning to decrease, and MAP with low oxygen and high carbon dioxide content increases the shelf-life by reducing ethylene biosynthesis and perception [94].

Some authors reported that MAP increases the fruit's bioactive compounds during storage (Table 2). Phenolic compounds are valuable for human health as antioxidants and

free-radical inhibitors, and affect the products' organoleptic properties [95]. In addition, ethylene synthesis is interrupted when oxygen is restricted and carbon dioxide increases, reducing the respiration rate [93]. When the respiration rate is reduced, the amount of carbohydrate consumed is also reduced, and the accumulated carbohydrates are used to produce phenolic compounds [96]. This increase in phenol content is due to genetic potential and environmental factors during growing and postharvest, such as a high $CO_2$ concentration that induces abiotic stress [93]. Some authors have also reported that high $CO_2$ content (5–7 kPa) prevents the oxidation of the main antioxidant compounds in fresh-cut strawberries [97]. On the other hand, carotenoid biosynthesis continues after fruit harvest and increases when exposed to a $CO_2$-rich modified atmosphere [93].

The carotenoids are related to vitamin A, and researchers believe that carotenoids absorb light wavelengths that cause chlorophyll optical oxidation, preserving chlorophyll and reducing damage [98]. Carotenoid oxidative degradation depends on environmental factors (oxygen level, temperature, and light) and the non-enzymatic oxidation of carotenoids by reactive oxygen [98,99]. Therefore, the increasing carotenoid content in MAP is probably due to the prevention of oxidative stress in the presence of high $CO_2$.

**Table 2.** Bioactive compounds in the fresh-cut fruits after the use of MAP during storage.

| Bioactive Compounds | MAP | Exposure Time | Result | Fresh-Cut Fruit | References |
|---|---|---|---|---|---|
| Vitamin C | 20% $CO_2$, in air | 10 days | 124 mg/L | Apple (braeburn) | [91] |
| Vitamin C | 7 Kpa $CO_2$ | 28 days | 5.9 mg/100 g | Apple (golden delicious) | [100] |
| Hydroxybenzoic acid | 2.5 $O_2$ + 7 Kpa $CO_2$ | 21 days | 10.1 mg/kg | Strawberry | [101] |
| p-Coumaric acid | 7 Kpa $CO_2$ | 21 days | 7.8 mg/kg | Strawberry | [101] |
| Ellagic acid | 7 Kpa $CO_2$ | 21 days | 73.8 mg/kg | Strawberry | [101] |
| Myricetin | 7 Kpa $CO_2$ | 5 days | 5.2 mg/kg | Strawberry | [101] |
| Quercetin | 7 Kpa $CO_2$ | 21 days | 33.5 mg/kg | Strawberry | [101] |
| Kaempferol | 7 Kpa $CO_2$ | 21 days | 4.0 mg/kg | Strawberry | [101] |
| Vitamin C | 7 Kpa $CO_2$ | 21 days | 400 mg/kg | Strawberry | [101] |

Several studies have shown physicochemical qualities such as a firmness to increase in MAP with high $CO_2$ levels of 10% to 20% during storage at 4 °C (e.g., for apple slices) [91]. Fruit softening occurs in the middle lamellar pectin degradation of cell walls due to cell wall hydrolase activity [102]. Furthermore, MAP induces the loss of tissue strength, depending on $O_2$ availability [91]. On the order hand, MAP treatment did not affect soluble solid content or total acidity during storage [91,103–105].

Postharvest handling of fresh-cut fruits such as apples and pears has been of concern due to contamination with different microorganisms. For example, fungal and total aerobic mesophilic bacteria counts on fresh-cut apples stored in MAP flushed with $CO_2$ (20%) decreased to 5 and 3.5 log CFU/g (control samples) until below 3.5 and 2.5 log CFU/g (MAP samples), respectively, on day 10 at 4 °C [91]. Furthermore, for fresh-cut pears, fungal and bacterial counts decreased from 5.4 and 6.5 CFU/g (control samples) to 4.0 and 6.0 CFU/g (21% $CO_2$), respectively, by 21 days at 4 °C [105].

The main pathogens found in strawberries are *Botrytis cinerea*, *Rhizopus* spp., *Mucor* spp., *Colletotrichum* spp., and *Penicillium* spp. [106]. The technique of MAP with a $CO_2$-rich atmosphere has been shown to effectively reduce the development of *B. cinerea*, *Rhizopus stolonifera*, and *Mucor* species in strawberry fruit [107,108]. For this reason, using MAP technology is one of the best methods and most studied for controlling fresh-cut fruits.

*3.2. Natural Preservatives*

3.2.1. Antioxidants

Besides peeling, cutting, or slicing, the loss of membrane integrity of the fruits is related to several stress factors that can generate reactive oxygen species (ROS). ROS can act as secondary messengers in various critical physiological phenomena; however, ROS also induce oxidative damage under stress conditions [109]. The ROS mainly comprise singlet

oxygen ($^1O_2$), hydrogen peroxide ($H_2O_2$), superoxide radicals ($O^{\bullet-}{}_2$), and hydroxyl radicals (OH$\bullet$). Degradation of pigments, lipids, proteins, and other biomolecules indicates cellular damage by ROS, which leads to cell death. To prevent or attenuate this phenomenon, exogenous antioxidants could be a helpful solution for fresh-cut fruits. These compounds inactivate ROS, PPO, and POX to reduce the formation of brown pigments [110]. Synthetic antioxidants have been used for this due to their high efficacy; however, consumers demand more natural and sustainable compounds as food ingredients.

Plant phenolic compounds are potent antioxidants due to their redox properties, reducing agents, hydrogen donators, and singlet oxygen quenchers [111]. Flavonoids are the main phenolic compounds of their entire class and are divided into flavonols, flavones, isoflavones, and anthocyanins. They have a role in scavenging principally $^1O_2$ [109]. Phenolic extracts from plants or fruits and their antioxidant potential to preserve fresh-cut fruits have been extensively studied (Table 3). For example, yerba mate, pomegranate, kiwifruit, elderberry flower, strawberry tree (leaves and branches), vine (leaves and branches), olive (leaves and branches), pear (pulp, peel, pomace), apple (peel and pomace), bitter melon, and mango seed extracts are used to reduce browning and increase the antioxidant capacities of fresh-cut mangoes, apples, and pears [110,112–115]. Some examples of phenolics from plant or fruit extracts are rosmarinic acid, p-coumaric acid, trans-cinnamic acid, hydroxyphenyl lactic acid, caffeic acid, gallic acid, vanillin, quercetin, resveratrol, and eugenol [116–118].

On the other hand, ascorbic acid is the most abundant and extensively studied antioxidant compound. Its mechanism of action is through electron donation to a wide range of enzymatic and non-enzymatic reactions. Ascorbic acid participates in the first defense against ROS attacks, reacting with $H_2O_2$, OH$\bullet$, and $O^{\bullet-}{}_2$; and it regenerates $\alpha$-tocopherol from tocopheroxyl radicals, protecting membranes from oxidative stress damage [109]. Table 3 shows some studies on reducing the browning of fresh-cut fruits, principally apples using ascorbic acid.

Another antioxidant commonly used in fresh-cut fruits is citric acid. This compound is often considered safe. It can prevent browning and fruit disease by reducing the respiration of postharvest fruits. Citric acid treatment can slow down the decreases in the soluble sugars and titratable acidity and is beneficial to maintaining fruit quality during storage [124]. Usually, this antioxidant compound affects PPO, and with its inhibition, produces an antibrowning effect on fresh-cut fruits, as is the case with apples [14,113,114].

Melatonin is a hormone from fruits or vegetables that can protect cell structure, reduce peroxide levels by removing free radicals, reduce lipid peroxidation, enhance oxidation resistance, and prevent DNA damage [120]. According to Dan et al. [125], this molecule also inhibits browning during the cultivation of plant tissue; therefore, it can inhibit the oxidation of phenolic compounds, being a powerful antioxidant. Studies in recent years have promoted fruit ripeness and improved its storage quality, as is the case for tomatoes [126] and strawberries [127]. They have also reduced surface browning on fresh-cut apples and pears [120]. Antioxidants such as natural preservatives are essential in the food industry because they allow foods to conserve their nutritional properties and quality levels, preventing contact between oxidative enzymes and phenols, and indirectly inhibiting enzymatic browning. However, limitations have been observed when applying antioxidants: for example, they could modify the natural flavor of fresh-cut fruits, which generally depends on the concentration—that is, the higher the concentration, the greater the susceptibility of the fresh-cut fruits to having their flavor modified [113,114]. This has the consequence that consumers reject fresh-cut fruits for consumption, as they do not perceive their natural flavor. Therefore, most studies focus on determining the optimal concentrations of antioxidants that preserve the nutritional and sensory quality of fresh-cut fruits.

**Table 3.** Effects of pure or extracted antioxidant compounds on the quality of fresh-cut fruits.

| Antioxidant/ Source of Antioxidant | Fresh-Cut Fruit | Treatment Application | Principal Results | References |
|---|---|---|---|---|
| Rosmarinic acid, p-Coumaric acid, Trans-Cinnamic acid, Hydroxyphenyllactic acid, caffeic acid, ascorbic acid, gallic acid, citric acid and BHA | Apple | Immersion (500 µg/mL of each antioxidant) | Reduced the browning, maintained the acidic pH and restricted growth of *L. monocytogenes* even after 10 days of treatment. | [116] |
| Calcium ascorbate; vanillin or cinnamic acid | Nectarine | Immersion (6% calcium L-ascorbate) | Reduced browning | [117] |
| Mango seed extract | Mango | Immersion (6.25 g/L of the extract) | Preserved fresh-cut fruits, increasing polyphenols, flavonoids and antioxidant capacity. | [112] |
| Apple polyphenols | Red pitaya | Spraying (5 g/L apple polyphenols) | Maintained sensory (retention of color, delay of the softening) and nutritional attributes of fresh-cut red pitaya fruit. | [119] |
| Yerba mate (*Ilex paraguariensis*) Citric acid Ascorbic acid | Apple | Infusion (1.2% yerba mate + 0.9% citric acid + 1.0% ascorbic acid) | Increased antioxidant capacity and decreased browning. The color, flavor and texture of the apples were kept. | [113,114] |
| Phenolics from juice or extract of pomegranate and kiwifruit | Pear | Immersion (0.3% of antioxidants) | Improved antioxidant capacity and prevented enzymatic browning. | [115] |
| Melatonin | Pear | Soaked with 0, 0.05, 0.1 and 0.5 mM melatonin | Reduced the surface browning, maintained the titratable acidity, enhanced total phenolic content and antioxidant capacity, and delayed the reduction of ascorbic acid. | [120] |
| Extracts of: Elderberry flower (*Sambucus* L.) Vine (*Vitis vinifera* L.) leaves and branches Pear (*Pyrus communis* L. "Rocha") pulp, peel and pomace Olive (*Oleo europaea* L.) leaves and branches Apple (*Malus domestica* L.) peel and pomace Acorn (*Quercus* L.) bark Bitter Melon (*Momordica charantia* L.) whole plant Strawberry tree (*Arbutus Unedo* L.) leaves and branches Potato plant (*Solanum tuberosum* L.) leaves | Pear | Sprayed (9.5 mg/mL, 5 mg/mL and 16 mg/mL) | Delaying fresh-cut pear browning expansion. Strawberry leaves and branches were the best antioxidant extracts. | [110] |
| Coconut liquid endosperm | Apple | Immersion (100% into the coconut liquid) | Coconut liquid endosperms are feasible natural agent inhibiting browning incidence of fresh-cut fruits during storage. | [121] |
| Melatonin | Apple and Pear | Immersion (0.05, 0.1, and 0.2 mM melatonin) | Reduced surface browning in fresh-cut foods. | [122] |
| Citric acid | Apple | Immersion (5% citric acid) | Inactivation of *Salmonella* and polyphenol oxidase. | [123] |
| Eugenol | Water chestnut | Immersion (0.4% and 1.5% eugenol) | Eugenol exhibited inhibitory effect on fresh-cut water chestnuts browning. Eugenol could also enhance the enzymatic/non-enzymatic antioxidant capacity and alleviate the ROS damage to membrane | [118] |

### 3.2.2. Antimicrobials

As was stated before, fresh-cut fruits are more susceptible to microbial spoilage than raw produce. Traditional antimicrobials have been widely used for many years to preserve food products. However, currently, there is great interest in using natural antimicrobial agents to prevent microbial deterioration, assure safe consumption, and retain the physicochemical and sensory quality of food due to the demand for healthier, fresher products with lower amounts of synthetic preservatives [128]. Emerging technologies involve using natural compounds with bioactive properties to enhance the shelf lives of fresh-cut fruits. They have been used against food spoilage and pathogenic microorganisms [129].

Several antimicrobials from various sources, such as animals (chitosan), plants (phenolic extracts, essential oils), and microbial (nisin), have been effectively used for the preservation of fresh-cut fruits [130]. The most common bioactive compounds with proven efficacy are plant derivatives such as essential oils and phenolic compounds [131]. Antimicrobials have different concentrations and effects for inhibition or inactivation of microbial growth. The specific antimicrobial targets include cell walls, membranes, metabolic enzymes, protein synthesis, and genetic systems. However, the exact antimicrobial mechanisms have not been fully elucidated. They also depend on the type, genus, species, and strain of the microorganism and environmental conditions (pH, water activity, temperature, and atmosphere, among others) [132].

Essential oils from spices and medicinal plants are sources of many bioactive compounds of great interest in the food industry with antimicrobial attributes [133–135]. Essential oils are oily liquids formed from complex mixtures of volatile compounds with strong aromas that are responsible for the fragrances of flowers and other plants. Thei main compounds are of low molecular weight, such as hydrocarbons, terpenes, terpenoids, and their derivatives [136]. Essential oils are produced by many plants as secondary metabolites and extracted from leaves, husks, bark, flowers, buds, and seeds through steam distillation, hydrodistillation, or solvent extraction [10]. Essential oils are used as flavorings and aromatized in perfumery, cosmetics, and the food industry because they are generally recognized as safe.

Numerous studies have shown the potential of essential oils against Gram-negative and Gram-positive bacteria and fungi. The antibacterial mechanisms of essential oils are related to their hydrophobicity and the structures of their components. The lipophilic nature of essential oils allows them to pass through the cell wall and damage the cytoplasmic membrane while disrupting various layers of polysaccharides, fatty acids, and phospholipids, eventually rendering them permeable. They can also bind to proteins to prevent them from carrying out their normal functions of transporting molecules and ions. Additionally, the hydrophobic components present in the essential oil could change the permeability of the microbial cell membrane for cations such as H+ and K+, which modify the flow of protons, changing the cellular pH and affecting the chemical compositions of the cells and their activity. Loss of differential permeability results in an imbalance in intracellular osmotic pressure, which then disrupts intracellular organelles, leading to the release of cytoplasmic contents, lower proton motive force, a depleted ATP pool, and the denaturing of various enzymes and proteins, ultimately causing cell death [137,138].

For decades, essential oils have been identified as food additives to preserve the microbiological quality of fresh-cut fruits. Emerging technologies are currently being studied to improve solubility and reduce essential oils' strong aromas and flavors, such as edible coatings and films added with natural antimicrobials. Essential oils can help decrease the permeability of water vapor in hydrophilic films due to their lipidic properties, and provide other helpful properties, such as structural, optical, tensile, and antimicrobial ones [139]. For example, hydroxypropyl methylcellulose coatings added with Thai essential oil were applied to mango to decrease the losses in weight and firmness, and color change. Additionally, this treatment showed antifungal activity against *Colletotrichum gloeosporioides* and good sensory acceptance [140].

A nanoemulsion of orange peel essential oil was added to pectin-based edible coatings at 0.5 and 1.0% to evaluate the quality of orange slices stored at 4 °C for 17 days. The slices coated with 1.0% essential oil showed fewer mesophilic aerobic microorganisms and yeasts and molds loads compared to samples without essential oil. The findings revealed that orange peel essential oil nanoemulsion-based edible coatings could improve the shelf life of orange slices without compromising sensory qualities [141]. In a similar approach, gellan coatings with added geraniol (1.2 and 2.4 μL) were used to preserve the quality of fresh-cut strawberries for 7 days at 5 °C. The coatings at both doses reduced mesophilic bacterium, yeast, mold, and psychrophilic counts compared to untreated samples. However, the treatments did not improve the texture and sensory quality; the fruit showed higher firmness loss than the control fruit [142].

Phenolic compounds are a large and varied group of molecules found in plants which have different structures and functions. These compounds are synthesized by the plant's secondary metabolism during the plant's normal development and once the plant tissue undergoes diverse types of stress [143]. According to their structural characteristics, about 10,000 phenolic compounds are divided into several groups. The molecular structure of any phenolic compound includes an aromatic ring attached to at least one hydroxyl group. Phenolic acids and flavonoids are the most investigated food additives among the several families of phenolic compounds [144]. Antimicrobial activity has been associated with hydroxyl groups' presence, number, and positions in their structures. The hydroxyl groups can bind to essential enzymes of microbial metabolism and increase the affinity for cytoplasmatic membranes, which can cause microbial death, inhibition of virulence factors, and biofilm formation [145]. It has been demonstrated that adding a single hydroxyl group and particular lipophilicity to a molecule increases its antibacterial capabilities [144].

Byproducts from fruits, such as peel and seed, have many phenolic compounds with antimicrobial potential. For example, pomegranate peel extract rich in phenolic compounds, such as ellagitannins, showed in vitro bactericidal and bacteriostatic activity against five strains of *L. monocytogenes*, and at 12 g/L demonstrated vigorous antibacterial activity against bacterial load on fresh-cut apple, melon, and pear throughout a 7-day storage period [128]. However, this study did not evaluate the sensory acceptance of the product, which is an essential part of the quality of fresh-cut fruits. It is essential to consider that the results of the in vitro activity are always better than those in vivo, since the presence of organic matter increases the survival of the bacteria and decreases the contact of the extracts with the bacteria. In addition, it should also be considered that the extracts with the highest content of phenolic compounds generally present the greatest effects and the highest pH, which influences the chemical structures and functions of polyphenols.

Similarly, mango seed extract was used at 6.25 g/L, the highest concentration with sensory acceptability, to preserve fresh-cut mango quality and reduce inoculated bacteria [112]. In the same approach, aqueous extracts of nut byproducts, cashew nut shell and coconut shell, were applied to fresh-cut papaya, thereby reducing the population of *E. coli*, *L. monocytogenes*, and *S. enterica*. However, cashew nut shell extract affected the sensory characteristics—darkening the fresh-cut papaya tissues. Therefore, the authors proposed coconut shell, rich in luteolin, as an excellent antimicrobial additive in food applications [146]. Grape byproducts, stem and leaf extracts, were micro-encapsulated and added as biopreservatives in a fresh-cut fruit salad of grape, apple, and sweet corn. The extract capsules formed a film on the fruits, reduced the inoculated loads of *Aspergillus ochraceus* and *Alternaria alternata*, and reduced ochratoxin, a dangerous mycotoxin. When applied to fresh-cut fruit salad, the stem capsules were more effective than leaf capsules [147].

Using natural compounds such as essential oils and plant extracts is quite promising for preserving fresh-cut fruits; however, their applications still face some challenges [148]. The application of natural antimicrobials presents some limitations due to their potent aromas, high reactivity, hydrophobicity, decreased solubility, and potential interactions with food's carbohydrates and fatty acids that could modify its organoleptic qualities [149]. For example, essential oils are highly irritating, which can cause damage to fruit tissues,

and have strong aromas and flavors that can be unpleasant for the consumer. Additionally, some plant extracts have strong colorations that can affect the color of food products [112]. In this sense, the optimal doses that are effective in inhibiting microbial growth and do not affect sensory quality should be evaluated. In addition, one must create a palatable combination with the aromas and flavor notes of essential oils or plant extracts and fresh-cut fruits. The appropriate combination might boost natural food preservation trends [10].

### 3.3. Physical Treatments

### 3.3.1. UV-C Radiation

Exploring more effective ways to maintain the quality of fresh-cut fruits has led us to look for other technologies, such as shortwave ultraviolet (UV-C, 190–280 nm) irradiation. This technique is approved by the Food and Drug Administration of the USA and is recognized as safe for food products because it is environmentally friendly and toxic-residue-free [150]. The uses that have been attributed to this technique in fresh-cut fruits are varied: inhibiting the browning process (reducing PPO), prolonging the shelf-life, maintaining optimal quality during storage, delaying ripening, reducing softening, and avoiding undesirable flavors [150–153]. Additionally, UV-C induces stress in fruit and consequently heightens the antioxidant defense system and some concentrations of secondary metabolites, such as phenolic compounds and pigments [154,155]. Additionally, UV-C radiation may extend the shelf life of fresh-cut products by its microbicidal effect due to the formation of pyrimidine dimers blocking the microbial cell replication due to DNA alteration [156].

A 4.54 kJ m$^{-2}$ dose was used for the UV-C treatment of fresh-cut pomegranate arils storage for 14 days. The end of shelf life showed the lowest mesophile, yeast, and mold growth and the highest total antioxidant activity and phenolic content [157]. Recently, fresh-cut lotus (*Nelumbo nucifera* Gaertn.) root exposed to a UV-C lamp (75 W) for 10 min and then stored for 8 days exhibited a significantly low browning degree, soluble quinone content, and inactivation of enzymes activities (polyphenol oxidase, peroxidase, and phenylalanine ammonia lyase) [152]. UV-C (4.5 kJ m$^{-2}$) disinfection treatment and nano-coating lemon essential oil nanocapsules were used to preserve fresh-cut cucumber for 15 days. A good correlation was observed between increasing the fruit's shelf life and decreasing its enzymatic activity [158]. Similarly, UV-C radiation (4.0 kJ m$^{-2}$) for 5 min combined with calcium lactate was applied to fresh-cut kiwifruit slices for 7 days of storage. UV-C and calcium lactate treatment could synergistically maintain overall quality and improve the antioxidant capacity of kiwifruit slices [153]. These studies demonstrate the usefulness of the maintenance and quality of fresh-cut fruits when applying UV-C radiation. However, the application of UV-C radiation does not have the same impact on all fresh cut fruits, which is a limiting factor for the application of this technology. For example, the application of UV-C in fresh-cut pomegranate arils decreased the amounts of yeast and mold, whereas in kiwi slices this decrease was not observed [153]. The exposure time is also an important factor, since being exposed for several minutes could cause damage to the tissues, causing nutritional and physicochemical changes. For this reason, an alternative use is to apply UV-C radiation with other technologies [153,158].

### 3.3.2. High Hydrostatic Pressure

High hydrostatic pressure (HHP) generates safe and stable food products and does not damage the sensory or nutritional properties of the product [159,160]. This technology requires high initial capital investment. However, the costs of the HHP-processed products have improved during the last few years [161]. In addition, HHP applications are increasing the health benefits to consumers by adding value products [162,163]. In general, HHP uses isostatic pressures between 100 and 1000 MPa and does not require heat [164]. HHP is a non-thermal process, and the temperature of water used does not exceed 50 °C [165]. HHP processing may present positive effects such as eliminating pathogenic and deteriorative microorganisms to ensure food safety and extend the product's shelf-life [166]. However,

HPP technology negatively affects food matrices by altering the structures of enzymes responsible for desired compounds due to the breakage of weak interactions (hydrophobic and electrostatic interactions) [167].

HHP has shown some potential applications in maintaining and enhancing nutritional value and bioactive compounds through microbial and enzyme inactivation, extending the shelf lives of different fruits [161]. However, various quality changes have been reported after using HHP, such as darkening of avocado slices and persimmon [168,169]. Denoya et al. [170] suggested that increased pressure in the HHP conditions decreases the activity of PPO in fresh-cut fruits such as peach, and found evidence of the close relationship between the PPO residual activity and the degree of browning for the processed peach pieces. However, it has been reported that increasing pressure in fresh-cut Hachiya persimmon causes changes in tissue structure and physicochemical properties [171].

On the other hand, HHP with moderate pressure (up to 600 MPa) for short holding times on colorimetric and textural parameters on fresh-cut pumpkins and peach had no effect, and the esterification degree of pectin decreased after HHP [170,172,173]. However, moderate pressures produce lysis of the cell membranes, causing the food matrix's mechanical disruption. To avoid this scenario, it is recommended to use 200 MPa to maintain the best physicochemical properties and preserve the integrity of the bioactive compounds [170,171]. These alterations could be because mechanisms of enzymatic inactivation by HHP are very complex and depend on the product and conditions [174].

HHP technology is cold pasteurization that inactivates microorganisms, significantly extending the products' shelf-life and guaranteeing food safety [161]. For example, HHP processing can ensure the microbiological safety of fresh-cut pumpkins. HHP immediately reduced the total plate counts (TPC) to 1.61 and 1.52 log10 CFU/g at 450 and 550 MPa, respectively, and to 2.57 and 1.69 log10 CFU/g, respectively, after 45 days [175]. This increase in the TPC could be because the HHP treatment does not entirely inactive microorganisms. In general, the tendency in pressure-treated products at a higher pressure and holding time treatments is to significantly increase mesophilic aerobic microorganisms' lag phase times [175]. This may be due to the greater severity of membrane structure damage, an increase in pH, and the recovery and growth of injured cells [176].

### 3.3.3. Ozone

Ozone has been widely used in the food and agriculture industry and is "generally recognized as safe" (GRAS) by the FDA (US Food and Drug Administration, 2001). The ozone treatments are mainly applied as aqueous ozone and gaseous ozone. Aqueous ozone, a sanitizing agent, is a highly effective disinfectant at low concentrations and short contact times, capable of rapidly decomposing into oxygen, and therefore, does not leave a residue on treated products [177,178]. Therefore, washing with ozone is a highly efficient and safe disinfection method to preserve fresh-cut fruits and vegetables and can be a good substitute for washing with chlorine. Recent studies have indicated that ozone can delay texture deterioration of fresh-cut fruits and vegetables such as apples, green bell peppers, and onions [179–181]. Gaseous ozone treatment is developed as a sanitization method to avoid cross-contamination when used with large volumes of produce that could not be sanitized by further washing with disinfectants [182].

Some studies evaluated the effectiveness of ozone in various fruits and vegetables. The effects of ozone are associated with multiple reactions, including the inactivation of enzymes, alterations in nucleic acid concentrations, and oxidation of membrane lipids [183]. Ozone treatment could reduce weight loss, minimize tissue destruction, maintain cell integrity, and thus reduce moisture transpiration [184]. Ozone has been praised for its efficacy in maintaining the firmness of products. However, the firmness of products inevitably decays over storage time. In general, ozone has made a minor impact on this reduction and does not affect the texture of fresh-cut products during the storage period [178,184].

The degradation of polyphenols during ozonation produces several chemical reactions [184]. This occurs due to the penetration of ozone into the cells with an oxygen ion that leads to direct responses with the compounds and the hydroxyl radicals produced through the oxygen ion catalyzed by the hydroxide ion [178,184,185]. Several authors have reported the main quality parameters of fresh-cut fruits treated with aqueous or gaseous ozone (Table 4). Aqueous and gaseous ozone treatments do not induce stress in the respiration rate in fresh-cut products and cause the inactivation of biosynthetic enzymes, which will be responsible for various metabolic activities, inclusive of ethylene biosynthesis for a brief time, and enhancing the antioxidant capacity and total phenolic content without compromising on its sensorial quality during storage [184,186].

**Table 4.** Quality parameters and microbiological control of fresh-cut fruits treated with aqueous or gaseous ozone.

| Fresh-Cut Fruit | Treatment | Results | References |
|---|---|---|---|
| Apple | Aqueous ozone 1.4 mg/L At 5 and 10 min | • Respiratory rate is maintained<br>• Ethylene production maintained at low levels<br>• PPO activity was decreased<br>• Causes the loss of antioxidant compounds<br>• Antioxidant capacity was increased | [184] |
| Apple | Aqueous ozone 1.4 mg/L at 5 min | • Softening during the storage is reduced<br>• Water-soluble pectin was increased<br>• Protopectin content, 4% KOH-soluble fraction, and cellulose content were decreased<br>• 24% KOH-soluble fraction was maintained<br>• Pectin methylesterase activity was increased<br>• β-galactosidase and α-arabinofuranosidase activities were inhibited<br>• Polygalacturonase activity was maintained | [179] |
| Papaya | Gaseous ozone 9.2 µL/L at 10, 20 and 30 min | • Total phenolic content was increased<br>• Ascorbic acid content was decreased | [187] |
| Durian | Gaseous ozone 900 mg/L at 3 and 5 min | • The appearance of the flesh and funiculus was maintained<br>• The respiration rate was reduced<br>• Ethylene production was reduced<br>• Total phenolic content was increased<br>• Antioxidant capacity was increased | [186] |
| Durian | Gaseous ozone 900 mg/L for 14 days at 4 °C of storage | • Total bacteria count was 2.10 log CFU/g<br>• The coliform population was 1.93 log CFU/g | [186] |
| Apple | Aqueous ozone 1.4 µL/L at 5 min for 12 days and 4 °C of storage | • Total bacteria count was 3.5 log CFU/g<br>• Total molds count was 1.42 log CFU/g<br>• Total yeasts count was 3.33 log CFU/g | [184] |
| Papaya | Gaseous ozone 9.2 µL/L for 10 and 30 min directly | • Total mesophilic bacteria value was 0.22 log CFU/g at 10 min<br>• Coliform count was 1.12 log CFU/g at 30 min | [187] |
| Bell pepper | Gaseous ozone 9 ppm for 6 h | • *E. coli* O157 count was 2.89 log CFU/g<br>• *S. Typhimurium* count was 2.56 log CFU/g<br>• *L. monocytogenes* count was 3.06 log CFU/g | [182] |

Ozone is a strong oxidant and an antimicrobial agent (Liu et al., 2016; Liu et al., 2021). The effectiveness of ozone in microorganisms increases proportionally to the increase

in concentration and exposure time [184,186]. Oxidative damage by ozone produces irregular cell structure in bacteria, inducing ROS destruction of lipid and protein molecules embedded in the bacterial cell membrane [182,188]. The lipid peroxidation by ozone forms lipid hydroperoxides (LOOH) that produce lipid degradation, leading to cell wall rupture, cellular leakage, excessive nutrient loss, and cell death [182]. Studies showed that aqueous and gaseous ozone could be employed to achieve desired microbial safety (Table 4). Ozone is an alternative sanitizer used for several fresh-cut fruits, achieving microbial reductions and increasing shelf life. However, ozone is a highly reactive compound. It may cause physiological injury to the product, and it is necessary to keep ozone concentrations as low as possible [182]. Some fresh-cut fruits and vegetables, such as apples, green bell peppers, and onions, are susceptible to ozone, causing texture deterioration [180,181,184]. The effectiveness of ozone depends on product type, $O_3$ concentration, and treatment time.

## 4. Futures Trends

Emerging technologies, such as natural antioxidant and antimicrobial compounds, modified atmosphere, ozone, UV-C, and high hydrostatic pressure, are still not fully implemented in the food industry because of their high costs, or they do not guarantee the total safety of the product. Therefore, future research should combine various preservation methods, known as hurdle technologies, to generate a synergistic effect due to the different mechanisms of action to increase food products' nutritional, sensory, and microbiological quality. They should also carry out studies optimizing these technologies for industrial applications to determine doses and secondary effects on the nutritional and sensory quality of the products. This will allow us to offer methods that produce fresher, safer, more nutritious, more natural food with fewer synthetic additives.

According to the information collected in this review, there are still many areas of opportunity in terms of developing and applying technologies such as pulsed light and smart packaging to study the increase in the quality of fresh-cut fruits. Some advantages and disadvantages of the different emerging technologies that can be applied to fresh-cut fruits were already seen in this review; however, one of these technologies that was not mentioned in the previous points is pulsed light treatments. Pulsed light treatment (intense broad spectrum ranging from UV to near-infrared) is a decontamination method for foods (inactivates microorganisms) consisting of repeatedly providing light at certain intervals during storage to further extend the shelf lives of fresh-cut fruits [189]. However, only a few studies have been conducted using this technology on fresh-cut fruits; for example, raspberries and blueberries [190], mangoes [191,192], apples [193], strawberries [194], and cantaloupes [189]. Therefore, this technology could be used on other fresh-cut fruits to increase their shelf lives while preserving their nutritional quality and sensory aspects. Other technologies that were not seen in this review were treatments with cold plasma, heat shock, and hydrogen sulfide, among others, which are very interesting technologies, but the information on their effects on fresh-cut fruits is very limited. For this reason, it was difficult to cover these issues in the current review.

Another significant area of opportunity is being able to complement emerging technologies using so-called smart packaging. Smart packaging is a broad concept that encompasses several functions related to packaged goods, such as foods and beverages, to maintain integrity and prevent food spoilage (shelf life) through visual or other indicators. In addition, they respond to changes in the environmental conditions of the product or the packaging. Finally, they must communicate the product's condition and report on the opening and integrity of the seal. Some smart packages are in use today, and many others are under development. However, most studies focus on whole fruits, and there are almost no studies on fresh-cut fruits [195,196], so this could be an excellent area of opportunity.

## 5. Conclusions

Fresh-cut fruits are in great demand for their convenience and nutritional quality. However, they are highly perishable and susceptible to contamination and microbiological

deterioration, which increases the risk of disease and compromises the nutritional and sensory quality of the product. The challenge for the food industries and the scientific community is the search for technologies to extend fresh-cut fruits' shelf lives. Emerging technologies are of recent creation and are alternatives to conventional technologies. They aim to eliminate microorganisms and maintain maximum nutritional and sensory quality. The objective, the nature of the fruit, its physiological characteristics, storage conditions, and efficacious doses, among other factors, must be considered to select the most suitable technology. This is because each fresh-cut fruit has distinctive characteristics, especially regarding bioactive compounds, since some products may contain them in greater or lesser amounts and even lack some of them. Within these technologies, active packaging, antioxidant and antimicrobial natural compounds, high hydrostatic pressure, UV-C radiation, and ozone are good candidates to extend the shelf lives of fresh-cut fruits and provide high microbiological and sensory quality during the preservation period.

**Author Contributions:** Writing-original draft preparation, A.T.B.-M., C.L.D.-T.-S. and R.D.I.-G.; conceptualization, O.M.-C., F.J.W.-C. and J.B.-F.; writing-review and editing, F.J.C.-M. and Y.I.C.-R.; supervision, S.R.-C. All authors have read and agreed to the published version of the manuscript.

**Funding:** This research received no external funding.

**Institutional Review Board Statement:** Not applicable.

**Informed Consent Statement:** Not applicable.

**Conflicts of Interest:** The authors declare no conflict of interest.

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
