# Peer review of "Emerging Technologies for Prolonging Fresh-Cut Fruits’ Quality and Safety during Storage"

_horticulturae, doi:10.3390/horticulturae8080731_

Round 1

Reviewer 1 Report

July 14, 2022

Comments entitled

Manuscript ID: horticulturae-1821378

Please see the  comments below.

General comment: extensive language editing is needed for this manuscript

                                   The authors should focus on fresh-cut fruit as the title

Specific comments:

Line 31. ‘Due to a population that is very busy with its different activities, and consequently with less time to prepare its food, ……’. What is ‘its’ represents in this sentence.

Section 2 title. I thought it would be smaller and straight forward

Effect of processing on the fresh-cut fruit quality and microbial safety: an overview

Line 127. ‘…..fresh-cut fruits are considered potentially dangerous products because

Line 137. however, these microorganisms cause the product to be dam aged and sensory unacceptable

Two types of microorganisms can contaminate these products, deteriorative and pathogenic. Generally, spoilage microorganisms do not cause serious harm to consumers, and pathogens are not responsible for fruit spoilage.

From Line 133-167, the information provided mainly focused on the type of microbes affects the fruit and the food born outbreak. It would have been better is the authors focused on how processing of fresh-cut fruit affects/leads to microbial contamination.

Table 1, it is not explained why the focus is only USA here. According to the introduction or the title of this review, its not specific to one country.

In think, use of terms should be consistent for example fresh-cut and minimally processed.

Line 147. I am not really sure if this sentence is correct, do you think they are perishable due to microbial contamination ……’Minimally processed fruits are highly perishable due to microbial contamination and 174 biochemical reactions that alter the food's sensory, safety, and physicochemical quality ….’ Please check it out. It is not necessary to repeat (information on Line 174-177) what has been said on the introduction section. I would think the authors shall start the section writing about edible coating

Line 177. ‘Films or coatings are considered any material used to wrap foods to prolong their shelf life and can be safely consumed with the food.’ `Do you mean any film can be consumed. Please consider writing the section

Line 180.  ‘They can also supply surface sterility and prevent losses of other important components’

Line 181 -185, please check the English language

Line 189. / line 195, I thought ‘This emerging technology c…’ should be ’These emerging technologies…

In the same line ‘This emerging technology can address customer demands for more natural, nutritious, ready-to-eat products, minimally processed, and high in nutritional content products without generating product waste’. Is there any difference between ‘nutritious’ and ‘high in nutritional content’

Line 195-197. Please reduce the sentence.

Line 217. How do ‘covering ’ can be explained as a deposition method

Line 222. Human variable?

Line 242. half-life of 242 minimally processed food?

Section 3.1.1 is missing information related to the limitations/problem of using edible coatings on fresh cut fruit

Line 308 ‘MAP is a method that involves the removal of air from inside the container and its 308 replacement by a gas or gas desire mixture (N2, CO2, and O2) supplied from pressurized 309 cylinders or otherwise; ….’ Be specific what type of MAP you are referring here.

Line 325-334. I am questioning the relevant or the proper presentation of this paragraph. The most important factors stated are oxidation, respiration, ethylene, ripening. However, these are not supported by any article on fresh-cut fruit, that if MAP particularly can assist to achieve reduction of these factors. Please considering adding supporting references otherwise its misleading, especially ripening and factors related to it in fresh-cut is not similar with fresh fruit

Line 337. ‘The high CO2 in cherimoya fruit increases the synthesis of phenylalanine 337 ammonia-lyase enzyme, which is the first phase in making phenolic compounds’..I don’t think this is relevant, this study is done on ripening cherimoya fruit. Please avoid such misleading information

Line 341.’ Free radicals produce senescence during the ripening process of fruits [105]’..free radicals cant produce senescence but the increase in free radicals could increase senescence. I thought it is irrelevant for the review content as well because this phenomena is expected during ripening of fruit  

Line 348. The first sentence is part of the previous paragraph.

Line 367-376. This main information in this section is written using whole fruit studies, I thought that would mislead a reader

Line 377-383. Again this is a whole fruit study report!. Please there are many similar writing styles in  this manuscript. The authors should amend those sections.

Table 3 is a summery of result from reference 89. May be add results from other studies as well

Line 385.’ The strawberry pathogens are B. cinerea, Rhizopus spp., Mucor spp., Colletotrichum 385 spp., and Penicillium spp.. please give a big emphasis on the consistency and writing. I thought you would like to say ‘’ The strawberry pathogens are B. cinerea, Rhizopus spp., Mucor spp., Colletotrichum spp., and Penicillium spp…’

Line 679. ‘Many emerging technologies are…..’ Is it not possible to list these technologies. Please consider listing them.

Line 688.’ According to the information collected in this review, there are still many areas of opportunity to study…’ what are these areas please

In the’ Future trends’ section briefly introduce the type of emerging techniques that are no part of the current review.  

Avoid discussion on the conclusion section

Author Response

We appreciate the time to review our manuscript and your positive feedback. The suggestions were addressed to improve the quality of our manuscript.

General comment: extensive language editing is needed for this manuscript.

Response: English was reviewed by an expert.

Specific comments:

Point 1. Line 31. 'Due to a population that is very busy with its different activities, and consequently with less time to prepare its food, ……'. What is 'its' represents in this sentence.

Response 1. Thank you for your observation. The sentence was changed "Due to a population that is very busy with their different activities, and consequently with less time to prepare their food…" English revision was done with an expert. Line 32.

Point 2. Section 2 title. I thought it would be smaller and straight forward.  Effect of processing on the fresh-cut fruit quality and microbial safety: an overview

Response 2. We appreciate the reviewer's comment, however our focus is more on emerging technologies on fresh-cut fruits during storage. In consultation with the authors of the article, we agreed to leave the current title.

Point 3. Line 127. '…..fresh-cut fruits are considered potentially dangerous products because

Response 3.  The sentence was removed.

Point 4. Line 137. however, these microorganisms cause the product to be dam aged and sensory unacceptable

Response 4.  The sentence was rewritten. Lines 126-130.

Point 5. Two types of microorganisms can contaminate these products, deteriorative and pathogenic. Generally, spoilage microorganisms do not cause serious harm to consumers, and pathogens are not responsible for fruit spoilage.

Response 5.  The sentence was changed in lines 140-141.

Point 6. From Line 133-167, the information provided mainly focused on the type of microbes affects the fruit and the food born outbreak. It would have been better is the authors focused on how processing of fresh-cut fruit affects/leads to microbial contamination.

Response 6.  Thank you for your suggestion. The sentences about foodborne outbreaks were reduced, and more information about how processing of fresh-cut fruit affects/leads to microbial contamination was added in lines124-137.

Point 7. Table 1, it is not explained why the focus is only USA here. According to the introduction or the title of this review, its not specific to one country.

Response 7.  The data was focused on the United States because this country's reports and monitoring are complete. However, this table was removed due to the previous suggestion.

Point 8. In think, use of terms should be consistent for example fresh-cut and minimally processed.

Response 8. We appreciate your comment. All terms were homogenized using fresh-cut fruits.

Point 9. Line 147. I am not really sure if this sentence is correct, do you think they are perishable due to microbial contamination ……' Minimally processed fruits are highly perishable due to microbial contamination and 174 biochemical reactions that alter the food's sensory, safety, and physicochemical quality ….' Please check it out. It is not necessary to repeat (information on Line 174-177) what has been said on the introduction section. I would think the authors shall start the section writing about edible coating

Response 9.  We agree with the reviewer. Fresh-cut fruits are not highly perishable due to microbial contamination and biochemical reactions. In fact, pre-cut fruits are considered perishable due to their intrinsic characteristics and minimal processing that favor microbial growth, which can cause changes in safety and sensory and physicochemical. The sentence was changed. The lines in 174-177 were removed according to the suggestion.

Point 10. Line 177. 'Films or coatings are considered any material used to wrap foods to prolong their shelf life and can be safely consumed with the food.' `Do you mean any film can be consumed. Please consider writing the section

Response 10. This was corrected, and the word edible was added to specify that any edible material can be used for this purpose (Line 205).

Point 11. Line 180.  'They can also supply surface sterility and prevent losses of other important components'

Response 11.  The sentence was removed.

Point 12. Line 181 -185, please check the English language

Response 12.  The sentence was rewritten 208-214. English was reviewed by an expert.

Point 13. Line 189. / line 195, I thought 'This emerging technology c…' should be' These emerging technologies…

Response 13.  The suggestion was addressed in line 217.

Point 14. In the same line 'This emerging technology can address customer demands for more natural, nutritious, ready-to-eat products, minimally processed, and high in nutritional content products without generating product waste'. Is there any difference between 'nutritious' and 'high in nutritional content'

Response 14.  Thank you for your observation. There is no difference between nutritious and high nutritional content. It was a mistake, and the sentence was corrected. Lines 218-219.

Point 15. Line 195-197. Please reduce the sentence.

Response 15.  We appreciate the suggestion; the sentence was split in two to make it shorter. Lines 223-227.

Point 16. Line 217. How do 'covering' can be explained as a deposition method

Response 16.  Thank you for your observation. The word covering was removed from the sentence. Line 245.

Point 17. Line 222. Human variable?

Response 17.  To avoid confusion, this part was modified. Line 250.

Point 18. Line 242. half-life of 242 minimally processed food?

Response 18.  The word half was changed by shelf in line 271.

Point 19. Section 3.1.1 is missing information related to the limitations/problem of using edible coatings on fresh cut fruit

Response 19.  We appreciate the suggestion; a paragraph considering the limitations of using edible coatings on fresh cut was added in lines 329-348.

Point 20. Line 308' MAP is a method that involves the removal of air from inside the container and its 308 replacement by a gas or gas desire mixture (N2, CO2, and O2) supplied from pressurized 309 cylinders or otherwise; ….' Be specific what type of MAP you are referring here.

Response 20.  The suggestion was addressed (Line 356-357).

Point 21. Line 325-334. I am questioning the relevant or the proper presentation of this paragraph. The most important factors stated are oxidation, respiration, ethylene, ripening. However, these are not supported by any article on fresh-cut fruit, that if MAP particularly can assist to achieve reduction of these factors. Please considering adding supporting references otherwise its misleading, especially ripening and factors related to it in fresh-cut is not similar with fresh fruit

Response 21.  We remove and add references that support the paragraph (Line 377-385).

Point 22. Line 337. 'The high CO2 in cherimoya fruit increases the synthesis of phenylalanine 337 ammonia-lyase enzyme, which is the first phase in making phenolic compounds'..I don't think this is relevant, this study is done on ripening cherimoya fruit. Please avoid such misleading information

Response 22.  The sentence was removed.

Point 23. Line 341.' Free radicals produce senescence during the ripening process of fruits [105]'..free radicals cant produce senescence but the increase in free radicals could increase senescence. I thought it is irrelevant for the review content as well because this phenomena is expected during ripening of fruit  

Response 23.  The sentence was removed.

Point 24. Line 348. The first sentence is part of the previous paragraph.

Response 24.  The sentence was moved to the previous paragraph (Line 394-396).

Point 25. Line 367-376. This main information in this section is written using whole fruit studies, I thought that would mislead a reader

Response 25.  The information was changed and added the correct information about fresh-cut fruit (Line 415-417).

Point 26. Line 377-383. Again this is a whole fruit study report!. Please there are many similar writing styles in this manuscript. The authors should amend those sections.

Response 26.  The information was changed and added correct information about fresh-cut fruit (Line 420-426).

Point 27. Table 3 is a summery of result from reference 89. May be add results from other studies as well

Response 27.  Now Table 3 is Table 2. New information was added in table 2.

Point 28. Line 385.' The strawberry pathogens are B. cinerea, Rhizopus spp., Mucor spp., Colletotrichum 385 spp., and Penicillium spp.. please give a big emphasis on the consistency and writing. I thought you would like to say" The strawberry pathogens are B. cinerea, Rhizopus spp., Mucor spp., Colletotrichum spp., and Penicillium spp…'

Response 28. The suggestion was addressed (Line 427-431).

Point 29. Line 679. 'Many emerging technologies are…..' Is it not possible to list these technologies. Please consider listing them.

Response 29. The suggestion was addressed in lines 749-750.

Point 30. Line 688.' According to the information collected in this review, there are still many areas of opportunity to study…' what are these areas please

Response 30. The suggestion was addressed in lines 760-761.

Point 31. In the' Future trends' section briefly introduce the type of emerging techniques that are no part of the current review.  

Response 31.  The suggestion was addressed in lines 771-774.

Point 32. Avoid discussion on the conclusion section

Response 32.  The suggestion was addressed.

Reviewer 2 Report

This is an interesting and well-written article that deals with a current topic of scientific interest and It is within the scope of this journal.
Minor points should be taken into consideration:
- Fresh-cut fruit quality is very dependent on refrigeration constant temperatures during the food chain, as the author knows. This issue is not discussed in the manuscript. Therefore, my recommendation is that some information should be added.
-The manuscript revised different technologies whose application promises to prolong fresh-cut quality, mainly the advantage. Could the authors add information about inconvenient or negative results examples, for example, UC-C, ozone, edible films /coatings added with natural oils herbs plants (antioxidants), or others?

Author Response

We appreciate the time to review our manuscript and your positive feedback. The suggestions were addressed to improve the quality of our manuscript.

-Point 1.  Fresh-cut fruit quality is very dependent on refrigeration constant temperatures during the food chain, as the author knows. This issue is not discussed in the manuscript. Therefore, my recommendation is that some information should be added.

Response 1. We added some information about the effect of refrigeration temperatures on fresh-cut fruits in lines 159-200.

Point 2. The manuscript revised different technologies whose application promises to prolong fresh-cut quality, mainly the advantage. Could the authors add information about inconvenient or negative results examples, for example, UC-C, ozone, edible films /coatings added with natural oils herbs plants (antioxidants), or others?

Response 2. Some limitations and inconvenience effects of emerging technologies were added in lines 328-347 (edible coatings), 488-495 (Antioxidants), 602-613 (natural antibacterial), 643-649 (UV-C), and 734-741 (ozone).

Round 2

Reviewer 1 Report

Thank you, authors, for revising the manuscript. The two important aspects  I thought the authors still have to consider as follows.

1. Introduction

Please, briefly summarized recently published reviews regarding to fresh cut fruit quality and safety and clearly explain how the current review is different from those one.

 2. References

Please remove references which are older, according to the aim, this review summarizes recent findings

Thank you 

Author Response

Thanks to the reviewer for their valuable comments and recommendations.

  1. Introduction. Please, briefly summarized recently published reviews regarding to fresh cut fruit quality and safety and clearly explain how the current review is different from those one.

Response 1. According to the reviewer's suggestion, we briefly described the most recent reviews in the area and highlighted the main differences compared to our manuscript, marked in green in the introduction. 

  1. References. Please remove references which are older, according to the aim, this review summarizes recent findings

Response 2. Most of the older references were deleted or updated (marked with green). We made an effort to leave the references from the previous five years. Still, some of them are necessary for the description of the manuscript or are the most recent information on the respective topic. We have approximately 85% of current references (2016 to date).